# Benzoquinoline Derivatives: An Attractive Approach to Newly Small Molecules with Anticancer Activity

**DOI:** 10.3390/ijms24098124

**Published:** 2023-05-01

**Authors:** Liliana Oniciuc, Dorina Amăriucăi-Mantu, Dumitrela Diaconu, Violeta Mangalagiu, Ramona Danac, Vasilichia Antoci, Ionel I. Mangalagiu

**Affiliations:** 1Faculty of Chemistry, Alexandru Ioan Cuza University of Iasi, 11 Carol 1st Bvd, 700506 Iasi, Romania; lili_oniciuc@yahoo.com (L.O.); dorina.mantu@uaic.ro (D.A.-M.); dumitrela.diaconu@uaic.ro (D.D.); rdanac@uaic.ro (R.D.); 2Institute of Interdisciplinary Research-CERNESIM Center, Alexandru Ioan Cuza University of Iasi, 11 Carol I, 700506 Iasi, Romania; violeta.mangalagiu@uaic.ro

**Keywords:** anticancer, benzo[*f*]quinoline derivatives, quaternary salts, cycloadducts

## Abstract

This study presents the synthesis, structural characterization, and in vitro evaluation of anticancer activity of some newly benzo[*f*]quinoline derivatives. The synthesis is facile and efficient, involving two steps: quaternization of nitrogen heterocycle followed by a [3+2] dipolar cycloaddition reaction. The synthesized compounds were characterized by FTIR, NMR, and X-ray diffraction on monocrystal in the case of compounds **6c** and **7c**. An in vitro single-dose anticancer assay of eighteen benzo[*f*]quinoline compounds, quaternary salts, and cycloadducts, was performed. The results showed that the most active compounds were quaternary salts **3d** and **3f** with aromatic R substituents. Quaternary salt **3d** revealed non-selective activity against all types of cancer cells, while salt **3f** exhibited a highly selective activity against leukemia cells. Compound **3d** also presented remarkable cytotoxic efficiency against four distinct types of cancer cells—namely, non-small cell lung cancer HOP–92, melanoma LOX IMVI, melanoma SK–MEL–5, and breast cancer MDA–MB–468. Compound **3f** was selected for five-dose screening. The study also includes SAR correlations.

## 1. Introduction

Benzo[*f*]quinoline is a polynuclear azaheterocycle with an extended π-π conjugation of potential interest in medicine, opto-electronics, agriculture, etc. [1,2]. Although the native heterocycle is known to be an environmental pollutant [3,4] and has been detected in coal tar [5], petroleum distillate [6], and urban air particles with genetic effects [2,7], recent research has shown that incorporating the benzo[*f*]quinoline skeleton into the design of certain compounds can be advantageous, resulting in new derivatives with biological properties such as antimicrobial [8,9,10] and anticancer activity [11,12,13]. Additionally, some derivatives have been designed and synthesized based on the fluorescent properties of benzo[*f*]quinoline [14,15], and some of these derivatives have been shown to have potential for use in organic light-emitting devices (OLED) [16,17,18,19]. In Figure 1, depictions of some of the potential applications of benzo[*f*]quinoline derivatives are shown.

Cancer is a highly lethal disease, with certain forms of the disease characterized by rapid progression and systemic destruction of the body [20]. According to the World Health Organization (WHO), cancer is the second leading cause of death globally, with over 10 million deaths recorded in 2020 [21]. The treatment of various types of cancer is difficult, being carried out surgically, either by radiotherapy or medication (chemotherapy), hormone therapy, immunotherapy, stem cell transplant, but also by a combination of some of them [22,23,24,25].

Chemotherapy is one of the anticancer therapies that can lead to an increase in life expectancy. In 1949, when the first agent with anticancer activity appeared on the market, life expectancy worldwide was 46.8 years; in 2015, when about 160 anticancer drugs were available for clinical use, life expectancy increased to 71.4 years [26,27]. However, the efficiency of the existing drugs from the market is low and there are many drawbacks. As a result, there is an urgent need for new anticancer drugs in therapy [28,29,30,31].

Considering the aforementioned factors, our research is aimed at developing novel compounds with anticancer properties by utilizing different basic scaffolds, including 8-aminoquinoline, 2-aminopyridine, imidazole/benzimidazole, pyridine, phenanthroline, phthalazine, or pyridazine, that could be potentially employed in chemotherapy for cancer treatment [32,33,34,35,36,37,38]. In this regard, we have diversified the core structure by incorporating benzo[*f*]quinoline, which has led to the synthesis of new hybrid compounds exhibiting anticancer activity.

## 2. Results and Discussion

### 2.1. Chemistry

In order to obtain the target benzo[*f*]quinoline derivatives (**HybB[f]Q**) with anticancer properties, we used a facile and efficient reaction pathway in two steps. The initial stage consisted of the quaternization reactions of benzo[*f*]quinoline **1** with variously reactive halides of type **2** (such as amides, esters, and aromatic ketones) when the corresponding quaternary salts **3a**–**g** were obtained (the synthesis of salts was reported elsewhere [8]). The second step consisted of a [3+2] dipolar cycloaddition reaction of ylides (generated in situ from the corresponding quaternary salts **3a**–**g**) with various dipolarophiles. In the subsequent step, the reaction conditions were optimized to achieve maximum yields of the targeted compound. The conditions were adapted based on the dipolarophile used, and involved the use of two distinct bases and solvents, respectively.

Consequently, the [3+2] dipolar cycloaddition reactions were first carried out to activate symmetrically substituted alkyne (dimethyl acetylenedicarboxylate (DMAD) **4a**) and nonsymmetrically substituted alkyne (ethyl propiolate **4b**), respectively, using quaternary salts **3a**–**c** as the starting materials. These reactions were performed in 1,2-butylene oxide, which was used both as a solvent and as a base.

In addition, [3+2] dipolar cycloaddition reactions were conducted to 1,4-naphthoquinone **5** (an activated symmetrically substituted alkene) in the case of quaternary salts with aliphatic salts **3a–c** and **3f,g**, respectively, with the solvent chloroform and by using triethylamine as a base. The cycloaddition reactions occurred in the desired pyrrolo-benzo[*f*]quinolines **6a-f** and isoindolo-benzo[*f*]quinolines **7a–e**, respectively. The synthetic route to obtain **HybB[f]Q** derivatives is illustrated in Figure 2.

The structures of newly **HybB[f]Q** compounds were proven by spectral analysis (FT–IR, HR–MS,^1^H–NMR, ^13^C–NMR, and two–dimensional experiments 2D–COSY, HMQC, HMBC), and single crystal X–ray diffraction in the case of compounds **6c** and **7c**. Table 1 displays the data obtained from the FT-IR and NMR spectral analyses.

In the ^1^H–NMR spectra of compounds **HybB[f]Q** with a pyrrolo–benzo[*f*]quinoline structure **6a–f**, the most deshielded signals are provided by the protons H–10 and H–11 from the benzene and pyridine rings, respectively. The chemical shift for H-10 signal is noted to be approximately 8.78 ppm in compounds **6a,b** with amide structures. For compounds **6c–f** with esteric structures, this signal is observed at chemical shifts with 0.22 ppm lower. The difference of approximately 0.23 ppm is also evident in the chemical shifts of the H-11 proton signals, which appears at 8.67 ppm for compounds **6a-b** and at 8.44 ppm for compounds **6c-f**, respectively. In the aliphatic region, the signals of the alkyl residues are observed. For cycloadducts **6a**, **6c**, and **6e**, formed by cycloaddition to DMAD, specific chemical shifts are observed for the signals furnished by the methyl H-15a (δ~ 3.94 ppm) and H-14a (δ~ 3.90 ppm). Similarly, for compounds **6b**, **6d**, and **6f**, obtained by the cycloaddition to EP, the signals from the ethyl residues appear at specific chemical shifts (δ~ 4.41 ppm for H-15 and δ~1.43 ppm for H-16).

In the ^1^H–NMR spectra of compounds **HybB[f]Q 7a–e** with an isoindolo–benzo[*f*]quinoline structure, the most downfielded signals belong to benzo[*f*]quinoline protons H–6, H–7 and H–8 within the chemical shift range of 8.80 ppm to 8.14 ppm.

The ^13^C-NMR spectra provide evidence for a completely aromatic structure in the case of **HybB[f]Q 6a–f** compounds, as indicated by the higher chemical shift values (ranging from 102.8 ppm to 139.4 ppm) of the carbon atoms in the pyrrolic cycle (C-1, C-2, C-3, and C-13). Likewise, for the cycloadducts **HybB[f]Q 7a–e**, the appearance of carbon atom signals (C-9, C-10, C-17, and C-19) from the pyrrolic cycle in the low-field region (as shown in Table 1) serves as evidence of a fully aromatic structure.

The FT-IR spectra of compounds **HybB[f]Q 6a–f** and **7a–e** reveal important absorption bands attributed to carbonyl groups. Specifically, carbonyl ester groups are characterized by absorption bands at 1722 cm^−1^ and 1690 cm^−1^ for **6a**, 1649 cm^−1^ for **6b**, 1716 cm^−1^ and 1693 cm^−1^ for **6c**, 1687 cm^−1^ for **6d**, 1700 cm^−1^ and 1690 cm-1 for **6e**, 1688 cm^−1^ for **6f**, 1723 cm^−1^ for **7b** and 1720 cm^−1^ for **7c**. For compounds **6a**, **6b**, and **7a**, which contain an amide group, the specific absorption band for the amide carbonyl group appears at around 1641 cm^−1^. The cyclic carbonyl ketones groups in **HybB[f]Q 7a-e** are identified by absorption bands ranging from 1673 cm^−1^ to 1661 cm^−1^ in the FT-IR spectra.

X-ray diffraction was performed on **HybB[f]Q** compounds **6c** and **7c** that yielded monocrystals, which were obtained as white or orange needles through crystallization from absolute ethanol.

Both **HybB[f]Q** compounds **6c** and **7c** were found to crystallize in an anhydrous form, with one molecule representing the asymmetric unit, as shown in Figure 1a,b, respectively. The structures of these derivatives were resolved using direct methods and refined in the P-1 space group type of the triclinic system. Compound **6c** is anhydrous and lacks typical intra- or inter-molecular interactions. In contrast, the molecules in compound **7c** are distributed in a planar configuration relative to each other, except for the ethyl carboxylic group(s), and do not exhibit typical inter-molecular interactions.

The Appendix A includes the NMR spectra of representative **HybB[f]Q** derivatives (**6b**, **6e**, **7b**, **7d**), as well as checkCIF files for the X-ray data of compounds **6c** and **7c**.

### 2.2. Anticancer Activity

The anticancer potency of the synthesized compounds was evaluated at the National Cancer Institute (NCI), USA, via their screening program for anticancer agents. This program, known as the NCI 60 cell line screen, is a valuable tool for drug discovery and development, as it provides a comprehensive assessment of the compounds’ activity against different types of cancer cells. The in vitro anticancer assay, using the NCI 60 cell line screening, includes 60 different human tumor cell lines representing various types of cancers, such as leukemia, melanoma, and cancers of the lung, colon, brain, ovary, breast, prostate, and kidney. The screening was performed in accordance with the NCI protocol [39,40,41].

The first step of the screening involves testing all selected compounds at a single dose of 10 μM against 60 cell lines [39]. The outcome of this single-dose screen is depicted as a mean graph, which is available for examination using the COMPARE program [42]. The results are expressed as the Percentage Growth Inhibition (PGI), indicating growth relative to the control with no drug and relative to the number of cells at time zero. This method allows for the identification of growth inhibition (values ranging from 0 to 100) as well as lethality (values less than 0). For instance, a value of 30 would indicate 70% growth inhibition, while a value of -30 would indicate 30% lethality.

Eighteen synthesized compounds, comprising seven **HybB[f]Q** salts **3a**–**g** and eleven **HybB[f]Q** cycloadducts (**6a**–**f**, **7a**–**e**), were subjected to a primary single-dose anticancer assay (at a concentration of 10^−5^ M). From the obtained data, the most active compounds were identified (the salts **3b**, **3d**, **3e**, **3f**, **3g**, the cycloadducts **6a**, **6d** (with pyrrolo-benzo[f]quinoline structure) and the cycloadducts **7b**, **7e** (with isoindolo-benzo[f]quinoline structure)), and the obtained results are listed in Table 2. The obtained results for all tested compounds can be found in the Appendix A (for salts **3a**–**g**) and Appendix A (for cycloadducts **6a**–**f** and **7a**–**e**).

Based on the results presented in Table 2, we noticed that the most effective compounds are **3d** and **3f**, which are quaternary salts containing an aromatic chain. Compound **3d** displays a nearly nonselective activity against all cancer cell types (with the exception of one ovarian cancer cell type and one renal cancer cell type), exhibiting excellent growth inhibition of 50–100% and significant lethality against four different cancer cell types (non-small cell lung cancer HOP-92, (8%), melanoma LOX IMVI (24%), SK-MEL-5 (89%) and breast cancer MDA-MB-468 (10%).

Quaternary salt **3f** demonstrates highly selective activity against leukemia, displaying a 100% growth inhibition and cytotoxicity against four different cell types: HL-60 (TB) (62% lethality); K-562 (26% lethality); MOLT-4 (3% lethality); SR, (28% lethality). Another noteworthy quaternary salt activity is **3e**, which exhibits a percentage growth inhibition of 80-100% against nine different cell types, including leukemia K-562 (PGI = 84%), non-small cell lung cancer HOP-92 (PGI = 88%), non-small cell lung cancer NCI-460 (PGI = 81%), colon cancer HT29 (PGI = 80%), melanoma SK-MEL-5 (PGI = 100%, L = 32%), melanoma UACC-257 (PGI = 85%), renal cancer SN12C (PGI = 83%), breast cancer MCF7 (PGI = 82%), and breast cancer MDA-MB-468 (PGI = 97%). The quaternary salt **3b** (with methyl esteric structure) is the most active in the aliphatic series, displaying good anticancer activity against the breast cancer MDA-MB-468 cell type (PGI = 85%). Among the cycloadducts, the most promising results are obtained with cycloadduct **6a** (with pyrrolo-benzo[*f*]quinoline structure), exhibiting a PGI of approximately 10-40% against the cancer cells.

The results in Table 2, as well as Appendix A, demonstrate that quaternary salts exhibit greater anticancer activity than cycloadducts. Considering the structure–activity relationship (SAR), certain observations could be carried out regarding the tested compounds:-Some compounds show high activity against multiple cancer cell lines, while others have a more selective effect;-The aromatic quaternary salts **3a**–**c** have better activity than aliphatic salts **3d**–**g**;-The major factor that affects the biological activity is the existence of the substituent from the para position of the benzoyl moiety. Thus, the compounds containing a methyl or a phenyl group exhibit the highest activity. Furthermore, the presence of a methoxy or chloro moiety seems to have a favorable effect on the activity;-The activity of quaternary salts is superior to that of **HybB[f]Q** cycloadducts, which could be attributed to the presence of a positively charged nitrogen atom in the molecule;-Cycloadducts of **HybB[f]Q** with a pyrrolo-benzo[f]quinoline structure exhibit greater activity than those with an isoindolo-benzo[f]quinoline structure. This suggests that a single fused cycle is preferable to two, for achieving anticancer activity.

It is noteworthy to mention that compound **3f** has been selected for the NCI 5-dose screening, which may provide valuable insights into its potential efficacy as an anticancer agent.

Taking into account our previous research [8,33,37,38], a hypothesis for the mechanism of action for quaternary salts may involve the interaction with ATP synthase or topoisomerase II. In addition, the ADMET parameters of the quaternary salts were previously presented [8], revealing a low toxicity profile for these compounds.

## 3. Materials and Methods

### 3.1. General Information

The reagents and solvents were purchased from commercial sources, being used without further purification. The melting points (uncorrected) of the new compounds were determined in open capillary tubes, using a MEL-TEMP (Barnstead International, Dubuque, IA, USA) Electrothermal apparatus. The nuclear magnetic experiments were recorded using two different spectrometers, Bruker Avance III 500 MHz/Bruker Avance DRX 400 MHz, operating at 500/400 MHz for ^1^H and 125/100 MHz for ^13^C nuclei (Bruker Vienna, Austria). Chemical shifts were reported in delta (δ) units (ppm), relative to the residual peak of solvents (ref: CDCl_3_/DMSO*-d6*, ^1^H: 7.26/2.50 ppm; ^13^C: 77.16/39.52 ppm) and coupling constants (*J*) in Hz. To describe the multiplicity of the ^1^H–NMR spectra, abbreviations were used: s = singlet, bs = broad singlet, as = apparent singlet, d = doublet, ad = apparent doublet, dd = doublet of doublets, add = apparent doublet of doublets, t = triplet, at = apparent triplet, td = triplet of doublets, atd = apparent triplet of doublets, dt = doublet of triplets, q = quartet, dq = doublet of quartets, m = multiplet. The IR spectra were recorded using a FTIR VERTEX 70 Bruker spectrometer (Bruker Optik, Leipzig, Germania) with an ATR module. The silica gel plates (Merck silica gel 60 F_254_ plates) were used for thin layer chromatography (TLC) visualization, carried out using a UV lamp (λ_max_ = 254 or 365 nm).

Single crystal X-ray diffraction analyses were carried out on a four-circle Rigaku Supernova dual Cu/Mo micro-focused source diffractometer. The apparatus was equipped with an EOS-CCD (charge-coupled device) detector with a cryo-system (OxfordCryosystem, London, UK), which allows samples to cool to −193.15 °C (80 K). The crystal samples were fixed onto the sample holder using a viscous oil-based cryoprotectant (Paratone^®^ N). The crystals were investigated at a temperature of 293 K using Cu X-ray radiation (CuKα = 1.5418 Å) with a 0.82 Å resolution limit. For data collection, cell refinement and data reduction were carried out using CrysAlisPro171.41.110a software from Rigaku OD. Structure determination, visualization, and analysis of the molecular crystal structure were carried out in Olex2 v1.3-ac4 software [43] using ShelXT 2018/2 [44] and ShelXL-2018/3 [45] to solve and refine the proposed structure models of the compounds through direct methods.

HR-MS experiments were recorded on a HESI-OrbitrapExploris 120 Mass Spectrometer (Thermo Fisher, Walthan, MA, USA) in positive mode.

### 3.2. General Procedure for the Synthesis of the HybB[f]Q **6a–f** with pyrrolo–benzo[f]quinoline Structure 

To a solution of benzo[*f*]quinolinium salts **3a–c** (0.5 mmol) in 15 mLof 1,2–butylene oxide, 1 mmol of dimethyl acetylenedicarboxylate (DMAD) **4a** or 1 mmol of ethyl propiolate **4b** was added. The mixture of reactions was refluxed during 3 h, and then stirred for 24 h at room temperature. The progress of the reactions was monitored by performing thin layer chromatography (TLC) using a mixture of CH_2_Cl_2_:CH_3_OH (5 mL:0.1 mL) as eluents. After the reaction was complete, the solvent was removed by means of concentration using a rotary evaporator, resulting in a crude oil. The desired **HybB[f]Q 6a-f** (Figure 3) were obtained by performing recrystallization from methanol.

#### 3.2.1. Dimethyl 3–carbamoylbenzo[*f*]pyrrolo[1,2–a]quinoline–1,2–dicarboxylate (**6a**)

Yellowish powder; yield: 60%; mp 254–256 °C; IR, ν_max_3410, 3301, 3189, 2960, 1722, 1690, 1644, 1468, 1224, 1101 cm^−1^; ^1^H NMR (500 MHz, DMSO-*d_6_*) δ 8.78 (1H, ad, J = 7.5 Hz, H–10), 8.67 (1H, ad, J = 9.0 Hz, H–11), 8.18 (6H, m, 2xNH, H–12, H–5, H–6, H–7), 7.77 (1H, at, J = 6.5 Hz, H–9), 7.70 (1H, ad, J = 8.0 Hz, H–8), 3.86 (6H, s, 3xH–14a, 3xH–15a); ^13^C NMR (125 MHz, DMSO-*d_6_*) δ 164.9 (C–15), 163.3 (C–16), 163.1 (C–14), 133.8 (C–13), 130.6 (C–4a), 130.2 (C–6a), 129.8 (C–5), 129.2 (C–10a), 128.6 (C–7), 128.0 (C–9),126.8 (C–8), 124.3 (C–3), 123.3 (C–10), 122.7 (C–2), 121.6 (C–11), 120.0 (C–10b), 117.4 (C–12), 116.6 (C–6), 102.8 (C–1), 52.4 (C–15a), 51.6 (C–14a); HESI–HRMS(+): m/z: calcd for [C_21_H_16_N_2_O_5_Na]^+^: 399.0956 [M+Na]^+^; found 399.0947.

#### 3.2.2. Ethyl 3–carbamoylbenzo[*f*]pyrrolo[1,2–a]quinoline–1–carboxylate (**6b**)

Cream powder; yield: 68%; mp 276–278 °C; IR, ν_max_3350, 3176, 2973, 2892, 1694, 1638, 1457, 1234, 1094cm^−1^; ^1^H NMR (400 MHz, DMSO-*d_6_*) δ 8.78 (1H, d, J = 8.40 Hz, H–10), 8.67 (1H, d, J = 9.6 Hz, H–11), 8.39 (1H, d, J = 9.6 Hz, H–12), 8.29 (1H, d, J = 9.2Hz, H–5),8.10 (2H, dd, J = 9.6 Hz, J = 8.0 Hz, H–6, H–7), 7.78 (1H, t, J = 7.6 Hz, H–9), 7.69 (4H, m, H–8, H–2, 2xNH), 4.38 (2H, q, J = 7.2 Hz, 2xH–15), 1.40 (3H, t, J = 7.2 Hz, 3xH–16); ^13^C NMR (100 MHz, DMSO-*d_6_*) δ 163.4 (C–17), 163.0 (C–14), 135.9 (C–13), 130.8 (C–4a), 129.8 (C–6a), 128.9 (C–10a),128.4 (C–7), 128.0 (C–6), 127.2 (C–9), 126.0 (C–8), 124.0 (C–3), 122.8 (C–10), 120.8 (C–11), 120.0 (C–2), 119.3 (C–10b), 118.1 (C–5), 117.0 (C–12), 104.5 (C–1), 59.0 (C–15), 13.9 (C–16); HESI–HRMS (+): m/z: calcd for [C_20_H_17_N_2_O_3_]^+^: 333.1239 [M+H]^+^; found 333.1225.

#### 3.2.3. Trimethyl benzo[*f*]pyrrolo[1,2–a]quinoline–1,2,3–tricarboxylate (**6c**)

Yellow powder; yield: 70%; mp 223–226 °C; IR, ν_max_2991, 2852, 1716, 1693, 1468, 1356, 1175, 1095 cm^−1^; ^1^H NMR (500 MHz, CDCl_3_) δ 8.53 (1H, d, J = 8.5 Hz, H–10), 8.47 (1H, d, J = 9.5 Hz, H–11), 8.39 (1H, d, J = 9.5 Hz, H–12), 7.97 (2H, m, H–5, H–6), 7.93 (1H, d, J = 7.5 Hz, H–7), 7.70 (1H, atd, J = 8.5 Hz, J = 8.0 Hz, H–9), 7.62 (1H, atd, J = 8.0 Hz, J = 7.5 Hz, H–8), 4.03 (3H, s, 3xH–15a), 3.98 (3H, s, 3xH–17), 3.94 (3H, s, 3xH–14a); ^13^C NMR (125 MHz, CDCl_3_) δ 166.3 (C–15), 163.4 (C–14), 161.2 (C–16), 138.0 (C–13), 132.0 (C–3), 131.6 (C–4a), 130.9 (C–6a), 129.6 (C–10a), 129.4 (C–6), 128.8 (C–7), 127.9 (C–9), 126.9 (C–8), 123.5 (C–11), 122.9 (C–10), 121.4 (C–10b), 118.8 (C–5), 117.7 (C–12), 117.0 (C–2), 104.5 (C–1), 53.0 (C–15a), 52.5 (C–17), 51.9 (C–14a); HESI–HRMS (+): m/z: calcd for [C_22_H_17_NO_6_K]^+^: 430.0692 [M+K]^+^; found 430.0680.

#### 3.2.4. 1–Ethyl 3–methyl benzo[*f*]pyrrolo[1,2–a]quinoline–1,3–dicarboxylate (**6d**)

Yellow powder; yield: 50%; mp 159–161 °C; IR, ν_max_2991, 2838, 1687, 1434, 1347, 1234, 1167, 1068 cm^−1^; ^1^H NMR (500 MHz, CDCl_3_) δ 8.54 (1H, d, J = 8.0 Hz, H–10), 8.44(2H, as, H–11, H–12), 8.25 (1H, d, J = 9.0 Hz, H–5), 8.04 (1H, s, H–2), 7.94 (2H, at, J = 10.0 Hz, J = 8.5 Hz, H–6, H–7), 7.68 (1H, t, J = 7.5 Hz, H–9), 7.60 (1H, t, J = 7.5 Hz, J = 7.0 Hz, H–8), 4.41 (2H, q, J = 7.5 Hz, 2xH–15), 1.45 (3H, t, J = 7.5 Hz, 3xH–16); ^13^C NMR (125 MHz, CDCl_3_) δ 164.2 (C–14), 162.3 (C–17), 139.5 (C–13), 132.0 (C–4a), 130.8 (C–6a), 129.7 (C–10a), 128.9 (C–7), 128.7 (C–6), 127.6 (C–9), 127.4 (C–2), 126.6 (C–8), 122.9 (C–10), 122.6 (C–11), 120.9 (C–10b), 119.4 (C–5),119.3 (C–3), 117.8 (C–12), 106.8 (C–1), 60.2 (C–15), 52.2 (C–18), 14.6 (C–16); HESI–HRMS (+): m/z: calcd for [C_21_H_17_NO_4_]^+^: 347.1157 [M]^+^; found 347.1145.

#### 3.2.5. 3–Ethyl 1,2–dimethyl benzo[*f*]pyrrolo[1,2–a]quinoline–1,2,3–tricarboxylate (**6e**)

Yellow powder; yield: 65%; mp 190–195 °C;IR, ν_max_ 3001, 2830, 1700, 1690, 1462, 1354, 1183, 1071 cm^−1^; ^1^H NMR (500 MHz, CDCl_3_) δ 8.52 (1H, d, J = 8.5 Hz, H–10), 8.47 (1H, d, J = 10.0 Hz, H–11), 8.39 (1H, d, J = 9.5 Hz, H–12), 7.99 (1H, d, J = 9.0 Hz, H–5), 7.94 (2H, m, H–6, H–7), 7.69 (1H, t, J = 7.5 Hz, H–9), 7.61 (1H, t,J = 7.5 Hz, H–8), 4.45 (2H, q, J = 7.5 Hz, 2xH–17), 4.03 (3H, s, 3xH–15a), 3.94 (3H, s, 3xH–14a), 1.42 (3H, t, J = 7.5 Hz, 3xH–18); ^13^C NMR (125 MHz, CDCl_3_) δ 166.3 (C–15), 163.5 (C–14), 160.8 (C–16), 137.9 (C–13), 131.9 (C–3), 131.6 (C–4a), 130.8 (C–6a), 129.5 (C–10a), 129.3 (C–7), 128.7 (C–6), 127.8 (C–9), 126.9 (C–8), 123.4 (C–11), 122.9 (C–10), 121.3 (C–10b), 119.0 (C–5), 117.7 (C–12), 117.2 (C–2), 104.5 (C–1), 61.8 (C–17), 52.9 (C–15a), 51.9 (C–14a), 14.22 (C–18); HESI–HRMS (+): m/z: calcd for [C_23_H_19_NO_6_]^+^: 405.1212 [M]^+^; found 405.1198.

#### 3.2.6. Diethyl benzo[*f*]pyrrolo[1,2–a]quinoline–1,3–dicarboxylate (**6f**)

Yellowish powder; yield: 55%; mp 179–182 °C; IR, ν_max_2994, 2840, 1688, 1430, 1352, 1233, 1169, 1053 cm^−1^; ^1^H NMR (500 MHz, CDCl_3_) δ 8.56 (1H, d, J = 8.5 Hz, H–10), 8.46 (2H, as, H–11, H–12), 8.26 (1H, d, J = 9.0 Hz, H–5), 8.05 (1H, s, H–2), 7.96 (2H, at, J = 10.0 Hz, J = 9.0 Hz, H–6, H–7), 7.69 (1H, atd, J = 7.0 Hz, J = 8.5 Hz, H–9), 7.61 (1H, atd, J = 8.0 Hz, J = 7.5 Hz, H–8), 4.44 (4H, dq, J = 7.0 Hz, 2xH–15, 2xH–18), 1.46 (6H, dt, J = 7.0 Hz, 3xH–16, 3xH–19); ^13^C NMR (125 MHz, CDCl_3_) δ 164.3 (C–14), 161.9 (C–17), 139.4 (C–13), 132.0 (C–4a), 130.8 (C–6a), 129.7 (C–10a), 128.9 (C–7), 128.7 (C–6), 127.6 (C–9), 127.2 (C–2),126.6 (C–8), 123.0 (C–10), 122.5 (C–11), 120.9 (C–10b), 119.8 (C–3), 119.5 (C–5), 117.8 (C–12), 106.8 (C–1), 61.2 (C–18), 60.2 (C–15), 14.7 (C–19), 14.5 (C–16); HESI–HRMS (+): m/z: calcd for [C_22_H_19_NO_4_]^+^: 361.1314 [M]^+^; found 361.1307.

### 3.3. General procedure for the synthesis of the HybB[f]Q **7a–e** with isoindolo–benzo[f]quinoline structure 

A solution of 1,4–napthoquinone **5** (0.55 mmol) and trimethylamine (0.75 mmol) in 15 mL of chloroform was added dropwise to a solution of benzo[*f*]quinolinium salts **3a–c** (0.5 mmol) or **3f,g** (0.5 mmol) in 15 mL chloroform. The obtained reaction mixtures were refluxed for 10 h, and then stirred for 48 h at room temperature. The completion of the reactions was monitored by performing thin layer chromatography (TLC) using a mixture of CH_2_Cl_2_: CH_3_OH (5 mL: 0.1 mL) as eluents. After completion of the reactions, the workup was found to be different depending on the rest (R) of **HybB[f]Q** with an isoindolo–benzo[*f*]quinoline structure **7a–e** (Figure 4). Thus, in case of **HybB[f]Q 7a–c** (R = amino or alkoxy moiety) the reaction mixtures were diluted with 40 mL of chloroform and then extracted three times with water (50 mL). The organic layers were dried over sodium sulphate, then filtered and evaporated. The obtained residues were crystallized from methanol. 

In the case of **HybB[f]Q 7d,e** (R=4-phenylphenyl or 4-chlorophenyl moiety), the obtained precipitates were filtered off. The purification was carried out by washing the precipitate with 100 mL of a mixture of hot solvents (CHCl_3_: CH_2_Cl_2_ = 9: 1, *v*:*v*).

Due to the poor solubility of these compounds in chloroform, in order to obtain suitable samples for the NMR experiments, the solutions of **HybB[f]Q 7a and 7c–e** in deuterated chloroform were treated with 1–2 drops of deuterated trifluoroacetic acid (TFA–*d*).

#### 3.3.1. 9,14–Dioxo–9,14-dihydrobenzo[*f*]benzo[5,6]isoindolo[2,1–a]quinoline–15–carboxamide (**7a**)

Dark orange powder; yield: 42%; mp > 300 °C; IR, ν_max_ 3381, 3206, 3076, 2990, 1663, 1639, 1460, 1243 cm^−1^; ^1^H NMR (500 MHz, CDCl_3,_ TFA–*d*) δ 8.69 (1H, d, J = 9.5 Hz, H–7), 8.60 (1H, d, J = 9.5 Hz, H–8), 8.53 (1H, d, J = 8.5 Hz, H–6), 8.19 (2H, atd, J = 7.5 Hz, J = 8.5 Hz, H–15, H–12), 8.86(3H, ad, J = 8.5 Hz, 2xNH, H–2), 8.00 (1H, d, J = 8.0 Hz, H–3),7.96 (1H, d, J = 9.0 Hz, H–1), 7.74 (4H, m, H–14, H–5, H–13, H–4); ^13^C NMR (125 MHz, CDCl_3,_ TFA-*d*) δ 183.2 (C–16), 181.1 (C–11), 167.4 (C–20), 137.9 (C–9), 135.4 (C–14), 134.9 (C–15a), 134.5 (C–11a), 134.4 (C–13), 131.4 (C–1a, C-6a), 130.1 (C–2), 129.4 (C–3), 129.1 (C–5), 128.8 (C–2a), 128.1 (C–12, C–4), 127.6 (C–7), 127.2 (C–15), 126.8 (C–18), 123.1 (C–6), 123.0 (C–6b), 121.8 (C–17), 118.1 (C–1), 117.8 (C–8), 112.2 (C–10); HESI–HRMS (+): m/z: calcd for [C_25_H_14_N_2_O_3_Na]^+^: 413.0902 [M+Na]^+^; found 413.0892. 

#### 3.3.2. Methyl 9,14–dioxo–9,14–dihydrobenzo[*f*]benzo[5,6]isoindolo[2,1–a]quinoline–15–carboxylate (**7b**)

Dark yellow powder; yield: 62%; mp > 300 °C; IR, ν_max_ 3047, 2945, 1723, 1673, 1670, 1548, 1466, 1242, 1213 cm^−1^; ^1^H NMR (500 MHz, CDCl_3_) δ 8.65 (1H, d, J = 9.5 Hz, H–7), 8.62 (1H, d, J = 8.5 Hz, H–6), 8.56 (1H, d, J = 9.5 Hz, H–8), 8.32 (1H, add, J = 9.0 Hz, J = 8.0 Hz, H–15), 8.25 (1H, add, J = 9.0 Hz, J = 7.5 Hz, H–12), 8.06(1H, d, J = 9.0 Hz, H–2), 7.98 (1H, d, J = 8.0 Hz, H–3), 7.93 (1H, d, J = 9.5 Hz, H–1), 7.77 (2H, t, J = 6.0 Hz, J = 7.5 Hz, H–5, H–14),7.72 (1H, atd, J = 7.5 Hz, H–13),7.67 (1H, t, J = 8.0 Hz, J = 7.0 Hz, H–4), 4.25 (1H, s, H–21); ^13^C NMR (125 MHz, CDCl_3_) δ 181.0 (C–11), 179.5 (C–16), 164.6 (C–20), 136.2 (C–15a), 134.9 (C–11a), 134.6 (C–9), 134.0 (C–14), 133,1 (C–13), 131.2 (C–2a), 131.1 (C–2), 131.1 (C–1a), 129.9 (C-6a),129.0 (C–3), 128.5 (C–5), 127.4 (C–4), 127.1 (C–12), 126.9 (C–15), 124.8 (C–18), 123.5 (C–8), 123.5 (C–6), 122.0 (C–6b), 120.3 (C–17), 119.0 (C–7), 115.5 (C–1), 111.6 (C–10), 54.1 (C–21); HESI–HRMS (+): m/z: calcd for [C_26_H_15_NO_4_Na]^+^: 428.0898 [M+Na]^+^; found 428.0890. 

#### 3.3.3. Ethyl 9,14–dioxo–9,14–dihydrobenzo[*f*]benzo[5,6]isoindolo[2,1–a]quinoline–15–carboxylate (**7c**)

Orange powder; yield: 61%; mp > 300 °C; IR, ν_max_3040, 2938, 1720, 1670, 1663, 1540, 1468, 1253, 1200 cm^−1^; ^1^H NMR (500 MHz, CDCl_3_, TFA–*d*) δ 8.31 (1H, d, J = 9.5 Hz, H–7), 8.24 (1H, d, J = 8.5 Hz, H–6), 8.14 (1H, d, J = 9.5 Hz, H–8), 8.02 (1H, d, J = 7.0 Hz, H–15), 7.96 (1H, d, J = 7.0 Hz, H–12), 7.90(1H, d, J = 9.5 Hz, H–2), 7.65 (4H, m, H–3, H–1, H–13, H–14), 7.44 (1H, t, J = 7.5 Hz, H–5), 7.30 (1H, t, J = 7.5 Hz, H–4), 4.75 (2H, q, J = 7.5 Hz, H–21), 1.55 (3H, t, J = 7.5 Hz, H–22); ^13^C NMR (125 MHz, CDCl_3_, TFA-*d*) δ 182.1 (C–16), 180.2 (C–11), 164.5 (C–20), 135.4 (C–9), 135.3 (C–11a), 135.0 (C–13), 134.2 (C–15a), 134.0 (C–14), 131.9 (C–2), 130.9 (C–1a), 130.7 (C–6b), 129.2 (C-6a), 128.8 (C–3), 128.7 (C–5), 127.6 (C–4, C–15), 127.2 (C–12), 125.4 (C–7), 123.9 (C–18), 122.8 (C–6), 121.8 (C–2a), 121.4 (C–17), 117.9(C–8), 114.9 (C–1), 110.5 (C–10), 64.8 (C–21), 13.7 (C–22); HESI–HRMS (+): m/z: calcd for [C_27_H_17_NO_4_K]^+^: 458.0794 [M+K]^+^; found 458.0781.

#### 3.3.4. 15–([1,1’–Biphenyl]–4–carbonyl)benzo[*f*]benzo[5,6]isoindolo[2,1–a]quinoline–9,14–dione (**7d**)

Orange powder; yield: 40%; mp> 300 °C; IR, ν_max_2980, 1661, 1633, 1545, 1464, 1243, 1192 cm^−1^; ^1^H NMR (500 MHz, CDCl_3_, TFA–*d*) δ 8.79 (1H, d, J = 9.5 Hz, H–7), 8.69 (1H, d, J = 9.5 Hz, H–8), 8.63 (1H, d, J = 8.5 Hz, H–6), 8.25 (1H, d, J = 7.5 Hz, H–15), 8.14 (2H, d, J = 8.0 Hz, 2xH–22), 8.01 (1H, d, J = 7.0 Hz, H–12), 7.92 (2H, t, J = 9.5 Hz, J = 9.0 Hz, H–2, H–4), 7.83 (1H, t, J = 7.5 Hz, H–14), 7.76 (5H, m, H–5, 2xH–23, H–1, H–13), 7.65 (1H, d, J = 7.5 Hz, H–3), 7.64 (2H, d, J = 7.5 Hz, 2xH–26), 7.44 (2H, t, J = 7.5 Hz, J = 7.0 Hz, 2xH–27), 7.43 (1H, t, J = 7.5 Hz, H–28); ^13^C NMR (125 MHz, CDCl_3_, TFA-*d*) δ 193.0 (C–20), 183.0 (C–11), 181.6 (C–16), 149.6 (C–24), 139.3 (C–25), 137.4 (C–9), 135.9 (C–15a), 135.7 (C–14), 134.7 (C–1a), 134.6 (C–13), 134.5 (C–11a), 133.6 (C–21), 132.6 (C–4), 131.3 (2xC–22, C–6b), 129.9 (C-2a), 129.3 (2xC–27, C–28), 129.2 (C–5, C–2), 128.5 (2xC–23), 128.3 (C–3), 128.2 (C–12), 128.0 (C–18), 127.7 (C–15), 127.6 (2xC–26), 126.8 (C–7), 125.7 (C–17), 123.2 (C–6), 122.8 (C–6a), 118.6 (C–8), 116.3 (C–1), 111.5 (C–10); HESI–HRMS (+): m/z: calcd for [C_37_H_21_NO_3_K]^+^: 566.1158 [M+K]^+^; found 566.1142.

#### 3.3.5. 15–(4–Chlorobenzoyl)benzo[*f*]benzo[5,6]isoindolo[2,1–a]quinoline–9,14–dione (**7e**)

Orange powder; yield: 45%;mp> 300 °C; IR, ν_max_2991, 1662, 1634, 1543, 1460, 1245, 1182 cm^−1^; ^1^H NMR (500 MHz, CDCl_3_, TFA–*d*) δ 8.80 (1H, d, J = 9.5 Hz, H–7), 8.71 (1H, d, J = 9.5 Hz, H–8), 8.65 (1H, d, J = 8.5 Hz, H–6), 8.27 (1H, d, J = 8.0 Hz, H–15), 8.07 (1H, d, J = 8.5 Hz, H–12), 7.94 (4H, m, 2xH–22, H–3, H–2), 7.83 (2H, m, H–14, H–5), 7.75 (1H, d, J = 7.5 Hz, H–13), 7.70 (3H, m, H–4, 2xH–23), 7.65 (1H, d, J = 9.0 Hz, H–1); ^13^C NMR (125 MHz, CDCl_3_, TFA-*d*) δ 191.6 (C–20), 182.6 (C–11), 181.6 (C–16), 137.4 (C–9), 135.8 (C–15a), 135,5 (C–14), 135.1 (C–21), 134.5 (C–13), 134.4 (C–11a), 133.3 (2xC–23), 132.4 (C–2), 132.2 (C–1a), 131.7 (2xC–22), 131.3 (C–6b), 131.2 (C–24), 129.9 (C-6a), 129.3 (C–3), 129.2 (C–5), 128.1 (C–4, C–12), 127.6 (C–15), 127.3 (C–18), 126.6 (C–7), 125.9 (C–17), 123.3 (C–6), 122.8 (C–2a), 118.6 (C–8), 116.1 (C–1), 111.6 (C–10); HESI–HRMS (+): m/z: calcd for [C_31_H_16_ClNO_3_Na]^+^: 508.0716 [M+Na]^+^; found 508.0703. 

### 3.4. Cell Proliferation Assay

The in vitro biological tests were performed by the National Cancer Institute (NCI, Bethesda, MD, USA), under the Developmental Therapeutics Program (DTP). 

For over half a century, the Developmental Therapeutics Program (DTP) at NCI has effectively led applications of late-stage preclinical drugs through the critical stages of development. Thus, this program has succeeded in the discovery and development of over 70% of the anticancer drugs used in current therapy [46].

The operation of this screen utilizes 60 different human tumor cell lines, representing leukemia, melanoma and cancers of the lung, colon, brain, ovary, breast, prostate, and kidney [47]. The aim is to prioritize for further evaluation synthetic compounds or natural product samples showing selective growth inhibition or cell killing of particular tumor cell lines. This screen is unique in that the complexity of a 60-cell line dose response produced by a given compound results in a biological response pattern which can be utilized in pattern recognition algorithms via COMPARE program [42].

The first screening step is the evaluation of all compounds against the 60 cell lines at a single dose of 10^–5^ M. The output from the single-dose screen is reported as a mean graph and is available for analysis using the COMPARE program.

#### The Standard NCI/DTP Methodology of the In Vitro Cancer Screen [48]

All 60 human tumor cell lines of cancer were grown in an RPMI (Roswell Park Memorial Institute) 1640 medium consisting of 5% fetal bovine serum and 2 × 10^−3^ M L-glutamine. For a usual screening test, the cancer cells are inoculated in 100 μL, and into 96 well microtiter plates (at different plating densities from 5000 to 40,000 cells/well based on the duplication time of specific cell lines). After the inoculation of cells, incubation of the microtiter plates is performed 24 h prior to the addition of experimental drugs, under specific conditions: 37° C, 5% of carbon dioxide, 95% of air and 100% of relative humidity. For each cell line, after 24 h, two plates are fixed in situ with TCA, to establish measurements of each cell population at the time of adding the testing compound (Tz). The testing compounds are solubilized in DMSO (dimethyl sulfoxide) at 400–fold the desired final maximum test concentration and stored in a freezer before use. After the addition of the testing compound, a part of the frozen concentrate is defrosted and diluted 2-fold compared to the final concentration, with a medium containing 50 µg/mL gentamicin. To the microtiter wells containing 100 µL of the medium, 100 µL of the testing compound is also added, resulting in the required final drug concentration (10^–5^ M). After adding the testing compound, the microtiter plates are incubated for another 48 h under the conditions previously reported. In the case of adherent cells, the test is finished by the addition of cold TCA. The cancer cells are fixed in situ by adding 50 µL of cold 50% (*w*/*v*) TCA (final concentration, 10% TCA), and then incubated at 4 °C for 60 min. The supernatant is discarded of, and the plates are washed 5 times with tap water and air dried. After drying, 100 µL of a fluorescent dye (sulforhodamine B) solution (0.4% (*w*/*v*) in 1% acetic acid) is added to each well, and plates are incubated at 25 °C for 10 min. The unbound fluorescent dye is removed by washing 5 times with an acetic acid solution of 1%, and then the plates are dried in air. The bound fluorescent dye is solubilized with a 10 × 10^−3^ M trizma base, and the absorbance is measured at a wavelength of 515 nm using an automated plate reader. The method is identical for the suspension cells, except that the test is completed by fixing settled cells at the base of the wells by adding 50 µL of TCA 80% solution (final concentration, 16% TCA). 

Using the seven absorbance measurements, the percentage growth inhibition (PGI) is calculated as: [(Ti – Tz)/(C – Tz)] × 100 (for concentrations for which Ti >/= Tz);

[(Ti – Tz)/Tz] × 100 (for concentrations for which Ti < Tz).

Where Tz = time zero, C = control growth, Ti = test growth in the presence of the drug at the concentration level.

For each testing compound, three dose–response parameters are calculated.

## 4. Conclusions

In summary, we reported the synthesis and anticancer activity of some newly benzo[*f*]quinoline derivatives. A two-step synthesis was used to obtain the compounds, involving a quaternization reaction followed by a [3+2] dipolar cycloaddition, resulting in eleven novel **HybB[f]Q** cycloadducts. The **HybB[f]Q** compounds were characterized by FTIR, NMR and X-ray diffraction with monocrystals. Eighteen benzo[*f*]quinoline compounds, including quaternary salts and cycloadducts, were selected and tested for anticancer properties by the NCI in a single-dose assay (10^–5^ M). Compound **3f** has been selected for a further five-dose evaluation of its potential as an anticancer agent. Two aromatic quaternary salts—**3d**, with an excellent PGI in the area of 50–100% and very good lethality against four different cancer cell types: Non-small Cell Lung Cancer HOP-92, Melanoma LOX IMVI, Melanoma SK-MEL-5 and Breast Cancer MDA-MB-468; and **3f**, showing high selectivity against leukemia HL-60 (TB), K-562, MOLT-4 and SR—were the most active compounds. The SAR correlations indicate that the presence of a positive nitrogen atom in the molecule and an aromatic R residue are favorable for anticancer activity. Furthermore, the substituent in the *para* position of the benzoyl residue is the major factor influencing this activity, with methyl and phenyl groups showing the most interesting activity. The synthesized compounds have promising anticancer activity, and quaternary salts **3d** and **3f** are potential candidates for further drug development.

## Data Availability

Not applicable.

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
