# Peer review of "Benzoquinoline Derivatives: An Attractive Approach to Newly Small Molecules with Anticancer Activity"

_ijms, 2023, doi:10.3390/ijms24098124_

Round 1

Reviewer 1 Report

The manuscript "Benzoquinoline derivatives: an attractive approach to newly small molecules with anticancer activity" by Oniciuc et al. is an interesting article presenting the synthesis and in vitro biological activity of eleven novel HybB[f]Q cycloadducts. The article is well written and organized, with promising results and interesting chemical approach but some aspects must be improved to increase the scientific soundness. Hence, the manuscript is suitable for publication after the following major revisions:

- Authors must clarify why their compounds didn't proceed to the NCI 5-dose screening and no IC50s are reported;

- Data or at least hypothesis about the MOA should be added;

- Citotoxicity in healthy cells should be reported;

- Few typo must be checked. For example: "the in vitro anticancer assay" (line 144) is spelled wrong; 

- Several literature references are missing. At the end of line 44, note 22 is not illustrative. Add more relevant papers describing the mentioned treatments (hormone therapy, immunotherapy, etc.).

- Similarly, at the end of line 50, authors didn't cite any articles to highlight the "urgent need for new anticancer drugs". Herein some significant papers in the field: Eur J Med Chem. Rep. 2021, 1, 100004, https://doi.org/10.1016/j.ejmcr.2021.100004; Bioorganic Chemistry 201876 , 314-325 https://doi.org/10.1016/j.bioorg.2017.12.013Eur J Med Chem. 2022 Dec 5;243:114744. doi: 10.1016/j.ejmech.2022.114744; Nanomedicine 2022 17:10717-739, https://doi.org/10.2217/nnm-2022-0010

Author Response

We would like to thank the reviewers who take the time to review our work and provide valuable suggestions to improve it.

Answers for Reviewer 1:

In the article the changes are done with “track changes”.

  1. “- Authors must clarify why their compounds didn't proceed to the NCI 5-dose screening and no IC50s are reported;”

Due to the lengthy testing process involved in NCI screening, we have chosen to focus our report solely on the one-dose screening results of the selected compounds. However, we have included in the manuscript that compound 3f has been chosen for the NCI 5-dose screening.

The changes in the article: in abstract lines 23-24; in anticancer activity lines 160-161; in conclusions lines 460-461.

  1. “- Data or at least hypothesis about the MOA should be added;”

Based on our previous research as referenced in [8, 33, 37, 38], we speculate that the mechanism of action for quaternary salts may involve the interaction with ATP synthase or topoisomerase II.

We added this information in the manuscript: in section anticancer activity lines 209-210.

  1. “- Citotoxicity in healthy cells should be reported;”

We added the next phrase in section anticancer activity lines 211-213: In addition, the ADMET parameters of the quaternary salts were previously presented [8], revealing a low toxicity profile for these compounds.

  1. “Few typo must be checked. For example: "the in vitro anticancer assay" (line 144) is spelled wrong;” 

Thank you for bringing this to our attention. We reviewed the manuscript and make the necessary corrections.

  1. “- Several literature references are missing. At the end of line 44, note 22 is not illustrative. Add more relevant papers describing the mentioned treatments (hormone therapy, immunotherapy, etc.).”

We introduced as suggested, the next references:

[23] Abraham, J.; Staffurth, J. Hormonal therapy for cancer. Medicine 2016, 44(1), 30–33, https://doi.org/10.1016/j.mpmed.2015.10.014.

[24] Esfahani, K.; Roudaia, L.; Buhlaiga, N.; Del Rincon, S.V.; Papneja, N.; Miller, W.H. Jr. A review of cancer immunotherapy: from the past, to the present, to the future. Curr Oncol. 2020, 27(Suppl 2), 87–97. doi: 10.3747/co.27.5223. [25] Czogała, W.; Czogała, M.; Kwiecinska, K.; Bik-Multanowski, M.; Tomasik, P.; Hałubiec, P.; Łazarczyk, A.; Miklusiak, K.; Skoczen, S. The Expression of Genes Related to Lipid Metabolism and Metabolic Disorders in Children before and after Hematopoietic Stem Cell Transplantation—A Prospective Observational Study. Cancers 2021, 13, 3614. https://doi.org/10.3390/ cancers13143614.

  1. - Similarly, at the end of line 50, authors didn't cite any articles to highlight the "urgent need for new anticancer drugs". Herein some significant papers in the field: Eur J Med Chem. Rep. 2021, 1, 100004, https://doi.org/10.1016/j.ejmcr.2021.100004; Bioorganic Chemistry2018, 76 , 314-325 https://doi.org/10.1016/j.bioorg.2017.12.013; Eur J Med Chem.2022 Dec 5;243:114744. doi: 10.1016/j.ejmech.2022.114744; Nanomedicine 2022 17:10, 717-739, https://doi.org/10.2217/nnm-2022-0010

We introduced as suggested, the next references:

[28] Barreca, M.; Spanò, V.; Raimondi, M. V.; Tarantelli, C.; Spriano, F.; Bertoni, F.; Barraja, P.; Montalbano, A. Recurrence of the oxazole motif in tubulin colchicine site inhibitors with anti-tumor activity. Eur. J Med Chem. Rep2021, 1, 100004, https://doi.org/10.1016/j.ejmcr.2021.100004; 

[29] El-Sherief, H.A.M.; Youssif, B.G.M.; Bukhari, S.N.A.; Abdel-Aziz, M.; Abdel-Rahman, H.M. Novel 1,2,4-triazole derivatives as potential anticancer agents: Design, synthesis, molecular docking and mechanistic studies. Bioorganic Chemistry 2018, 76, 314–325, https://doi.org/10.1016/j.bioorg.2017.12.013; 

[30] Barreca, M.; Spanò, V.; Rocca, R.; Bivacqua, R.; Abel, A.C.; Maruca, A.; Montalbano, A.; Raimondi, M.V.; Tarantelli, C.; Gaudio, E.; Cascione, L.; Rinaldi, A.; Bai, R.; Steinmetz, M.O.; Prota, A.E.; Alcaro, S.; Hamel, E.; Bertoni, F.; Barraja, P. Development of [1,2]oxazoloisoindoles tubulin polymerization inhibitors: Further chemical modifications and potential therapeutic effects against lymphomas. Eur J Med Chem. 2022, 243, 114744. doi: 10.1016/j.ejmech.2022.114744.

[31] Vieira, A.M.G.; Silvestre, O.F.; Silva, B.F.B.; Ferreira, C.J.O.; Lopes, I.; Gomes, A.C.; Espiña, B.; Sárria, M.P. pH-sensitive nanoliposomes for passive and CXCR-4-mediated marine yessotoxin delivery for cancer therapy. Nanomedicine 2022, 17(10), 717-739, https://doi.org/10.2217/nnm-2022-0010

Reviewer 2 Report

In this manuscript the authors report the synthesis, structural determination and characterization of a series of new benzoquinoline derivatives and data on their anticancer activity.

The introduction is clear and give references on the potential applications of benzo[f]quinolines as well as the classical need for the development of new chemotherapy tools against cancer. The last paragraph from lines 60 to 64 is redundant to the former one and should be deleted.

The chemical part is well presented and appears to be sound. Scheme 2 clearly presents the synthetic route to obtain all new compounds. The part devoted to the structural analyses is also very clear, nevertheless data presented in Table 1 are also redundant to the ones listed for each compound in the Materials and Methods section, so such Table can also be deleted.

The anticancer activity of the compounds has been determined, thanks to the NCI screening program, at only one concentration of 10-5 M which is a quite high concentration to give valuable insights for the possible development of such compounds in oncopharmacotherapy. Some IC50s determined for the for active compounds remain awaited.

In the Materials and Methods section, a large part of the text is only a total copy/paste of paragraphs available from references 39 and 41 (lines 404-407 and much more from lines 419 – 452). This is not allowable for such a scientific manuscript. Line 594, indication of reference 41 is missing and the URL must be corrected.

Author Response

We would like to thank the reviewers who take the time to review our work and provide valuable suggestions to improve it.

Answers for Reviewer 2:

In the article the changes are done with “track changes”.

  1. “The introduction is clear and give references on the potential applications of benzo[f]quinolines as well as the classical need for the development of new chemotherapy tools against cancer. The last paragraph from lines 60 to 64 is redundant to the former one and should be deleted.”

We deleted the lines 60-64.

  1. “The chemical part is well presented and appears to be sound. Scheme 2 clearly presents the synthetic route to obtain all new compounds. The part devoted to the structural analyses is also very clear, nevertheless data presented in Table 1 are also redundant to the ones listed for each compound in the Materials and Methods section, so such Table can also be deleted.”

We included this information in a summarized form in Table 1 to support the discussions from the Results and discussion chapter, to facilitate the reader's comprehension by providing a quick overview of the key spectral data of the compounds.

  1. “The anticancer activity of the compounds has been determined, thanks to the NCI screening program, at only one concentration of 10-5 M which is a quite high concentration to give valuable insights for the possible development of such compounds in oncopharmacotherapy. Some IC50s determined for the for active compounds remain awaited.”

Due to the lengthy testing process involved in NCI screening, we have chosen to focus our report solely on the one-dose screening results of the selected compounds. However, we have included in the manuscript that compound 3f has been chosen for the NCI 5-dose screening.

The changes in the article: in abstract lines 23-24; in anticancer activity lines 211-213; in conclusions lines 517-518.

  1. “In the Materials and Methods section, a large part of the text is only a total copy/paste of paragraphs available from references 39 and 41 (lines 404-407 and much more from lines 419 – 452). This is not allowable for such a scientific manuscript. Line 594, indication of reference 41 is missing and the URL must be corrected.”

We rewrote the lines 423-425:

“The Developmental Therapeutics Program (DTP) from NCI, over half a century, has leading favorably the applicants for late–stage preclinical drug by the ending steps of progress to first–in–human studies. Thus, this program succeeded in the discovery and development of over 70% of the anticancer drugs used in current therapy [46].”

We rewrote the lines 442 – 475:

“All the 60 human tumor cell lines of the cancer are grown in RPMI (Roswell Park Memorial Institute) 1640 medium consisting of 5% fetal bovine serum and 2x10-3M L-glutamine. For a usually screening test, the cancer cells are inoculated in 100 μL, into 96 well microtiter plates (at different plating densities from 5,000 to 40,000 cells/well based on the duplication time of specific cell lines). After inoculations of cells, is done the incubation of the microtiter plates, for 24 h before to adding of experimental drugs, under special conditions: 37° C, 5 % of carbon dioxide, 95 % of air and 100 % of relative humidity. From each cell line, after a time of 24 hours, two plates are fixed in situ with TCA, to establish a measuring of the each cell population at the time of adding the testing compound (Tz). The testing compounds are solubilized in DMSO (dimethyl sulfoxide) at 400–fold the wished final maximum test concentration, stored frozen before to use. After the addition of testing compound, a part of the frozen concentrate is defrost and diluted to 2-fold compared to the final concentration, with a medium containing 50 µg/ml gentamicin. To the microtiter wells containing 100 µL of medium, also are added amounts of 100 µL of the testing compound, resulting in the required final drug concentration (10–5 M). After adding the testing compound, the microtiter plates are incubated for another 48 hours under the special conditions previously reported. In case of adherent cells, the test is finished by the addition of cold TCA. The cancer cells are fixed in situ by adding of 50 µL of cold 50 % (w/v) TCA (final concentration, 10 % TCA), and then incubated at 4 °C during 60 minutes. The supernatant is throw away, and the plates are washed 5 times with tap water and dried in air. After drying to each well is added 100 µL of a fluorescent dye (sulforhodamine B) solution (0.4 % (w/v) in 1 % acetic acid), and plates are incubated at 25 °C during 10 minutes. The unbound fluorescent dye is removed by washing 5 times with acetic acid solution 1 %, and then the plates are dried in air. The bound fluorescent dye is solubilized with 10x10-3M trizma base, and the absorbance is measuring at a wavelength of 515 nm using an automated plate reader. The method is identical for the suspension cells, except that the test is completed by fixing settled cells at the basis of the wells by adding 50 µL of TCA 80 % solution (final concentration, 16 % TCA).

Using the seven absorbance measurements the percentage growth inhibition (PGI) is calculated as: [(Ti–Tz)/(C–Tz)] x 100 (for concentrations for which Ti>/=Tz);

[(Ti–Tz)/Tz] x 100 (for concentrations for which Ti<Tz).

Were Tz = time zero, C = control growth, Ti = test growth in the presence of drug at the concentration level.

For each testing compound are calculated three dose response parameters.”

  1. “Line 594, indication of reference 41 is missing and the URL must be corrected.”

We have updated our manuscript by adding new bibliographic notes, including a reference to source 41, which has now been renumbered as reference 48. Additionally, we have made corrections to the URL.

Round 2

Reviewer 1 Report

Authors made the required changes. The article is now suitable for publication.

Author Response

Thank you very much.

Reviewer 2 Report

The authors have revised their manuscript taking into account the comments that some parts were redundant and large parts of the former version were copy/paste from already available documents.

Nevertheless, the biological part remains quite weak with no more precise data for the compounds that are presented as the most valuable. In order to assess any relevant activity, the data must be more precise such as, for example and at least for some new compounds, IC50s determined on one or several cancer cell lines. Such data remain awaited.

Author Response

  1. Nevertheless, the biological part remains quite weak with no more precise data for the compounds that are presented as the most valuable. In order to assess any relevant activity, the data must be more precise such as, for example and at least for some new compounds, IC50s determined on one or several cancer cell lines. Such data remain awaited.

Answer:

We appreciate the importance of having precise data to support the biological activity of the compounds we presented.

As we stipulated in manuscript, compound 3f was selected for NCI’ five dose testing, but we are unable to predict when those results will be available (usually it is a lengthy process).

Obtaining IC50 values depends of the availability of resources, such as cancer cell lines, reagents, and equipment, which may vary depending on the laboratory and institution, and unfortunately for us, it is impossible to obtain in a short time.

Given these constraints, we believe that the data we provided in the revised submission are still relevant and valuable for the scientific community. They demonstrate the potential of our compounds to inhibit the growth of human cancer cells and provide insights into their structure-activity relationship.

Round 3

Reviewer 2 Report

The authors do not present additional data in this third revised version, as it was also the case in the second revised version.

No additional comment in reply to the authors response.